# The Big Five as Predictors of Cognitive Function in Individuals with Bipolar Disorder

**DOI:** 10.3390/brainsci13050773

**Published:** 2023-05-08

**Authors:** Eva Fleischmann, Nina Dalkner, Frederike T. Fellendorf, Susanne A. Bengesser, Melanie Lenger, Armin Birner, Robert Queissner, Martina Platzer, Adelina Tmava-Berisha, Alexander Maget, Jolana Wagner-Skacel, Tatjana Stross, Franziska Schmiedhofer, Stefan Smolle, Annamaria Painold, Eva Z. Reininghaus

**Affiliations:** 1Department of Psychiatry and Psychotherapeutic Medicine, Medical University of Graz, 8036 Graz, Styria, Austria; eva.fleischmann@medunigraz.at (E.F.); frederike.fellendorf@medunigraz.at (F.T.F.); susanne.bengesser@medunigraz.at (S.A.B.); melanie.lenger@medunigraz.at (M.L.); armin.birner@medunigraz.at (A.B.); robert.queissner@medunigraz.at (R.Q.); martina.platzer@medunigraz.at (M.P.); adelina.tmava@medunigraz.at (A.T.-B.); alexander.maget@medunigraz.at (A.M.); tatjana.stross@medunigraz.at (T.S.); franziska.schmiedhofer@medunigraz.at (F.S.); stefan.smolle@medunigraz.at (S.S.); annamaria.painold@medunigraz.at (A.P.); eva.reininghaus@medunigraz.at (E.Z.R.); 2Department of Medical Psychology and Psychotherapy, Medical University of Graz, 8036 Graz, Styria, Austria; jolana.wagner-skacel@medunigraz.at

**Keywords:** bipolar disorder, personality, Big Five, cognitive function

## Abstract

The connection between cognitive function and the “Big Five” personality traits (openness, conscientiousness, extraversion, agreeableness, and neuroticism) in the general population is well known; however, studies researching bipolar disorder (BD) are scarce. Therefore, this study aimed to investigate the Big Five as predictors of executive function, verbal memory, attention, and processing speed in euthymic individuals with BD (cross-sectional: n = 129, including time point t1; longitudinal: n = 35, including t1 and t2). Participants completed the NEO Five-Factor Inventory, the Color and Word Interference Test, the Trail Making Test, the d2 Test of Attention Revised, and the California Verbal Learning Test. The results showed a significant negative correlation between executive function and neuroticism at t1. Changes in cognitive function between t1 and t2 did not correlate with and could not be predicted by the Big Five at t1. Additionally, worse executive function at t2 was predicted by higher neuroticism and lower conscientiousness at t1, and high neuroticism was a predictor of worse verbal memory at t2. The Big Five might not strongly impact cognitive function over short periods; however, they are significant predictors of cognitive function. Future studies should include a higher number of participants and more time in between points of measurement.

## 1. Introduction

Bipolar disorder (BD) is a severe mental disorder affecting 1–2% of the population worldwide. Characterized by depressive and (hypo-)manic episodes, the disorder typically manifests in young adulthood during the emergence of a stable structure of personality [1]. Individuals with BD often suffer from social problems, such as lower social skills [2] and lower perceived social support, than healthy controls (HC) [3], and experience stigma [4]. In addition, cognitive impairment is frequently found in individuals with BD [5,6,7], even before illness onset [8] and during times of euthymia [9].

Cognitive dysfunction has been found to worsen in concert with illness duration [10] as well as neuroprogression [11], although results suggesting no apparent link with the latter show the questionability of this issue [12]. The primarily affected domains of cognitive impairment are verbal memory [13,14], executive function [14], attention [14,15], and psychomotor processing speed [16], which represent three of the six domains of cognition described in the Diagnostic and Statistical Manual of Mental Disorders (DSM)-V [17]. Moreover, the fourth domain of social cognition can be impaired as well [18], specifically emotion recognition [19] and emotion regulation in social situations [20].

Correlates and predictors of cognitive impairment include psychotic symptoms and mood symptoms [21], especially depressive symptomatology [22] and a high number of manic episodes [23]. Furthermore, it has been suggested that psychopharmaceuticals might impede cognitive function as well, among them being antipsychotics and sodium valproate [24]. In daily life, cognitive function is negatively associated with work performance [9], sleep disturbance [25], and quality of life [26]. In addition, associations with personality were found, which will be explained in the following paragraphs.

The five-factor model (FFM; [27]), also known as the “Big Five” [28], includes five personality dimensions that remain stable over time: openness (O), conscientiousness (C), extraversion (E), agreeableness (A), and neuroticism (N). In comparison with HC, individuals with BD score higher in N [29,30]. N is associated with the occurrence of depressive and hypomanic symptoms [31], subjective well-being, insomnia, and anxiety [32]. C was reported to be lower in BD [29,30]. E was found to be lower by some studies [29,33], while genotype analysis has shown associations with both lower [30] and higher values of E [34]. E can, if strongly pronounced, predict the future onset of BD after a period of 14 years [35], as well as symptoms of hypomania [31], and is positively associated with social impairment [36]. According to genotype analysis, A might be lower in BD [30]. Low A was found to indicate a higher risk of developing (hypo)mania in individuals with depression and anxiety [37]. O was found to be more highly pronounced in BD, and seems to be associated with creative achievement [38].

In recent years, several studies have explored the relationship between these facets of personality and cognitive function in HC. Research has emphasized high N as one of the most significant personality predictors of impaired cognition [39,40,41]. According to Stephan [42], this might be explained by correlates of N impacting long-term cognitive function, such as high stress sensitivity [43], sleep disturbances [44], and alcohol use [45]. Low O is an important factor as well [42,46], as high O is associated with more activities that are cognitively stimulating [47]. Studies analyzing neural correlates of the FFM further support the notion that high N and low O impact cognition, finding relationships between worse white matter integrity and high N as well as between better white matter integrity and high O [48]. Additionally, low C might contribute to cognitive decline [46], as affected individuals display less behavior contributing to the preservation of cognitive function [49]. Lastly, low E is associated with lower verbal fluency, and findings for A are not consistent throughout the literature [50]. Table 1 shows a summary of the relationships between each of the Big Five and cognitive function in HC.

Concerning individuals with BD, two studies found high O to be a predictor of better cognitive function [38,51]. In particular, O’s association with ideas and values was the most important facet, correlating with several cognitive factor scores, such as auditory memory, emotional processing, verbal fluency, and processing speed [51]. Furthermore, a higher number of single-nucleotide polymorphisms of the brain-expressed protocadherin 17 gene showed correlations with both impaired cognition and higher N. Increased gene expression was found in individuals with BD compared to HCs [52]. Another study suggested a negative correlation between N and reaction time in the Affective Go/No-Go paradigm with a bias towards affective stimuli, which was not significant in HC and might suggest a greater receptivity to emotional stimuli in individuals with BD [53].

As there is currently a lack of studies investigating personality and cognitive function in BD, the aims of this study were 1. to investigate cross-sectional correlations between the Big Five as well as executive function, verbal memory, attention, and processing speed, and 2. to predict cognitive decline according to the Big Five in a longitudinal sample. It was hypothesized that high O, C, and E, as well as low N, would correlate with and predict better cognitive function, while analyses concerning A would be explorative.

## 2. Materials and Methods

### 2.1. Participants and Procedure

All participants were participants in the ongoing BIPFAT/BIPLONG study, which has aimed to investigate BD in a longitudinal setting since 2012. Conducted at the outpatient center for BD at the Medical University of Graz, Austria, Department of Psychiatry and Psychotherapeutic Medicine, the study’s focal points are lifestyle, lipid metabolism, inflammation processes, cognition, and brain function. Inclusion criteria were a BD diagnosis by trained specialists using the Structured Clinical Interview for DSM-IV [54], age between 18 and 70 years, and IQ of ≥80 at the time of measurement. Euthymia was defined in this study by a score of ≤12 on the Young Mania Rating Scale (YMRS) [55] and a score of ≤10 on the Hamilton Depression Scale (HAMD) [56]. Individuals were excluded if they suffered from severe immunological disorders, organic brain diseases, or dementia.

As cross-sectional and longitudinal comparisons were made, the study included two samples, with some participants being included in both. The cross-sectional sample, consisting of 129 individuals with BD, included data from the participants’ first visit (t1). The longitudinal sample of 35 individuals with BD additionally included assessments from a second visit yielding complete datasets (t2), taking place 345 days to 2106 days after the first visit. This study was conducted in accordance with the Declaration of Helsinki as well as approved by the Ethics Committee of the Medical University of Graz (EK-number: 25-335 ex 12/13), and all patients signed informed consent forms before participation.

### 2.2. Measurements

Participants completed a cognitive assessment as well as a personality questionnaire, all of which were administered in German. In addition, clinical and sociodemographic data were assessed by interview.

#### 2.2.1. Neuropsychological Assessment

Verbal memory was measured with the California Verbal Learning Test (CVLT) [57]. In particular, recall trials 1-5, short delay free recall, short delay cued recall, long delay free recall, and long delay cued recall were measured.

To assess attention and processing speed, three tests were used: the Trail Making Test part A (TMT-A) [58], the d2 Test of Attention Revised (d2-R) [59], and the word-reading and color-naming trials from the Color and Word Interference Test by J. R. Stroop [60].

Cognitive flexibility, one aspect of executive function, was assessed by the Trail Making Test part B (TMT-B) [58], as well as the interference trial from the Color and Word Interference Test by J. R. Stroop [60].

#### 2.2.2. Personality Assessment

The NEO Five-Factor Inventory (NEO-FFI) [61] was used to assess the five personality dimensions of the FFM. Participants were asked to rate 60 questions on a five-point Likert-type scale, with 1 = *strongly disagree* and 5 = *strongly agree*. For each factor, the sum score of the 15 corresponding questions is calculated, ranging from 15 to 75, with a higher score indicating a stronger expression of the respective factor. The NEO-FFI shows good internal consistency, with Cronbach’s alpha ranging between .63 and .82 for each factor [62].

### 2.3. Statistical Methods

For the cross-sectional sample, cognitive test scores were converted into z-scores and then summed up into three scores measuring three cognitive domains: (1) attention and processing speed (d2-R, Stroop color naming, Stroop word reading, TMT-A); (2) verbal learning and memory (CVLT trial 1-5, CVLT short delay free recall, CVLT long delay free recall, CVLT short delay cued recall, CVLT long delay cued recall); and (3) executive function (Stoop interference and TMT-B). Measures expressing reaction times were reversed before calculating the domain scores, because in contrast to the other scores, lower scores indicated higher performance. A higher domain score indicated higher performance.

For the longitudinal sample, the same procedure was repeated; however, not all cognitive test scores could be used for the creation of the domain scores, as they were not assessed at t2. For memory, only the score for recall trials 1-5 was included in the sum score. The sum score of attention and processing speed did not include the d2-R test, which was accounted for when calculating the sum score and comparing both samples. There were no differences in the calculation of executive function.

Mean Big Five values of the cross-sectional sample were compared to the norm sample using summary t-tests. To compare the t1 and t2 data of the longitudinal sample, t-tests were used for age and the HAMD, and a Wilcoxon Test was used for YMRS due to outliers. A repeated-measures multivariate analysis of variance (MANCOVA) was employed to test for differences in cognition between both time points. Only the variables assessed at both time points were included: TMT-A, TMT-B, recall trials 1-5 of the CVLT, and the Stroop test. Covariates included age, sex, education, time difference between t1 and t2, BDI, and illness duration. Key assumptions of the repeated-measures MANCOVAs (linearity and normality) were tested graphically, as well as with the Kolmogorov–Smirnov test.

Cross-sectional partial correlation analyses between each of the Big Five and verbal memory, executive function, attention, and processing speed were performed at time t1. Age, sex, education, BDI, and illness duration were used as covariates. A false discovery rate (FDR) was used to correct for alpha error cumulation. Furthermore, the same analyses were performed for the purpose of correlating each of the Big Five assessed at t1 with cognition at t1 and t2, as well as the difference between the three cognitive functions measured at t2 as compared to t1. In addition to the previously mentioned covariates, the time difference between t1 and t2 was used for the latter analysis.

Three multiple hierarchical regression analyses to predict executive function, verbal memory, attention, and processing speed at t2 were conducted. The first step included the variables of age, sex, education, time difference between t1 and t2, BDI, and illness duration, while the second step included the Big Five at t1. Three more hierarchical regression analyses to predict changes in cognitive function were calculated with the same variables. Conditions for multiple regression analyses were tested by using correlations (linearity), scatterplots (homoscedasticity), histograms (normal distribution of error variance), Durbin–Watson tests (lack of autocorrelations), and both variance inflation factor and tolerance (lack of multicollinearity).

The current study included participants who completed all relevant questionnaires. In sum, 12 individuals had to be excluded in the cross-sectional sample and 5 in the longitudinal sample due to missing data.

A post-hoc sensitivity analysis performed with G*Power [63] showed that a correlation coefficient with 129 participants would be sensitive to effects of *r* = .24 with 80% power (Correlation ρ H_0_ = 0, power = 0.80, α = 0.05, two-tailed). For regression analyses with 35 participants, the threshold of sensitivity was .48 (H_0_ ρ^2^ = 0, power = 0.80, α = 0.05, number of predictors = 11, two-tailed).

## 3. Results

### 3.1. Sample Characteristics

Sociodemographic information of the cross-sectional sample (n = 129) is displayed in Table 2, and of the longitudinal sample (n = 35) in Table 3. The Big Five scores of the cross-sectional sample were compared to the norm sample (*n* = 871) [61] with *t*-tests, and it was found that N was higher (*M* = 20.99, *SD* = 7.89, *t*(998) = 8.45, *p* < 0.001), while E (*M* = 26.88, *SD* = 6.47, *t*(998) = −2.35, *p* = 0.019) as well as C (*M* = 32.61, *SD* = 6.11, *t*(153,965) = −2.16, *p* = 0.032) were lower in individuals with BD. There was no difference in O (*M* = 29.47, *SD* = 6.53, *t*(998) = −0.07, *p* = 0.948) or A (*M* = 30.45, *SD* = 5.38, *t*(156,218) = −0.12, *p* = 0.906).

*T*-tests showed that both groups of the longitudinal sample differed in age, but not HAMD (see Table 3). A Wilcoxon test further resulted in no differences between the groups regarding YMRS. A repeated-measures MANCOVA with the covariates of age, sex, education, BDI, time difference between t1 and t2, and illness duration regarding cognition was not significant (F(6,19) = 0.68, *p* = 0.666).

### 3.2. Partial Correlation Analyses

Partial correlation analyses including the three parameters of cognitive function; the Big Five; and the covariates of age, sex, education, BDI, illness duration, and time difference between t1 and t2 for the longitudinal sample are shown for both samples in Table 4 and Table 5, respectively. In the cross-sectional sample, only executive function correlated negatively with N after correction with FDR. In contrast, the positive correlation between attention, processing speed, and O did not remain significant. In the longitudinal sample, partial correlation analyses with the same variables were conducted for both time points t1 and t2, and for the cognitive function differences between t1 and t2. N at t1 correlated significantly with memory (r = −0.52, *p* = 0.007) and executive function at t2 (r = −0.43, *p* = 0.034); however, neither correlation remained significant after the usage of FDR. Correlations between cognitive function differences and the Big Five did not yield any significant results.

### 3.3. Multiple Hierarchical Regression Analyses

#### 3.3.1. Executive Function

A multiple hierarchical regression analysis was performed to predict executive function at t2 (see Table 6). The variables of age, sex, education, time difference between t1 and t2, BDI, and illness duration entered in the first step did not yield a significant result (R^2^ = 0.38, R^2^ corr. = 0.23, F(6,24) = 2.46, *p* = 0.053). The second step, comprising the Big Five at t1, was significant (R^2^ = 0.65, R^2^ corr. = 0.45, F(11,19) = 3.20, *p* = 0.013). The results were significant for the predictors of education, time difference between t1 and t2, C, and N.

A second multiple hierarchical regression analysis predicting the change in executive function between t1 and t2, with the variables of age, sex, education, time difference between t1 and t2, BDI, and illness duration entered in the first step and the Big Five at t2 entered in the second step, was not significant (Model 1: R^2^ = 0.26, R^2^ corr. = 0.07, F(6,24) = 1.39, *p* = 0.260; Model 2: R^2^ = 0.37, R^2^ corr. = 0.01, F(11,19) = 1.03, *p* = 0.463).

#### 3.3.2. Verbal Memory

A multiple hierarchical regression analysis predicting verbal memory at t2 with the predictors of age, sex, education, time difference between t1 and t2, BDI, and illness duration showed a significant first step (R^2^ = 0.45, R^2^ corr. = 0.31, F(6,24) = 3.27, *p* = 0.017). Age was a significant predictor (see Table 7). The second step, including the Big Five at t1, was significant as well (R^2^ = 0.68, R^2^ corr. = 0.49, F(11,19) = 3.61, *p* = 0.007), showing significant effects of age and N.

A second multiple hierarchical regression analysis with the variables of age, sex, education, BDI, illness duration, and time difference between t1 and t2 entered in the first step was conducted to predict the change in verbal memory between t1 and t2. The Big Five were entered in the second step, and neither model yielded significant results (Model 1: R^2^ = 0.13, R^2^ corr. = −0.09, F(6,24) = 0.60, *p* = 0.729; Model 2: R^2^ = 0.40, R^2^ corr. = 0.06, F(11,19) = 1.17, *p* = 0.368).

#### 3.3.3. Attention and Processing Speed

A multiple hierarchical regression analysis predicting attention and processing speed at t2 with the variables of age, sex, education, time difference between t1 and t2, BDI, and illness duration as predictors in the first step was significant (R^2^ = 0.52, R^2^ corr. = 0.40, F(6,24) = 4.26, *p* = 0.005). The Big Five at t1 were included in the second step, which yielded significant results (R^2^ = 0.60, R^2^ corr. = 0.43, F(11,19) = 2.84, *p* = 0.022). Age was the sole significant predictor in the second step (see Table 8).

Another multiple hierarchical regression analysis, with the predictors of attention and processing speed difference between t1 and t2, was conducted. The first step included age, sex, education, time difference between t1 and t2, BDI, and illness duration, and the second step included the Big Five at t1, with neither model showing significant results (Model 1: R^2^ = 0.34, R^2^ corr. = 0.17, F(6,24) = 2.04, *p* = 0.099; Model 2: R^2^ = 0.41, R^2^ corr. = 0.07, F(11,19) = 1.21, *p* = 0.344).

## 4. Discussion

A cross-sectional (n = 129) and a longitudinal (n = 35) subsample of euthymic individuals with BD were investigated concerning the association between cognitive function and the Big Five. A significant negative correlation between executive function and neuroticism at t1 was found. Partial correlations and regression analyses predicting changes in cognitive function between t1 and t2 with the Big Five did not yield any significant results. High neuroticism at t1 was a significant predictor of worse executive function and verbal memory at t2, while high conscientiousness at t1 significantly predicted worse executive function at t2.

N was higher, and E, as well as C, were lower in the cross-sectional BD sample than in the norm sample [61], which is consistent with other studies [29,33]. Similarly, one study found this FFI triad with the opposite configuration to be a strong correlate of positive mental health [57], showing individuals with BD to be naturally vulnerable. Findings suggesting higher O [30,38,64] and lower A in BD [30] could not be replicated, and should be investigated in more detail.

Interestingly, variables of the Big Five at t1 predicted executive function and verbal memory at t2, but not the change in cognitive function between t1 and t2 in the longitudinal sample. It should be mentioned that mean cognitive test scores did not differ between t1 and t2. This lack of significant results might indicate that while certain aspects of personality are important for cognition, the influence exerted is only observable over a longer period of time, or that it is stronger during a different stage of development. Gale et al. [65] proposed that an association between personality and cognitive function found in mid-life might be a reflection of a correlation between these traits in childhood. It should be further considered that any meaningful associations between the Big Five and cognition might have been underestimated as a consequence of the small sample sizes.

Only the negative correlation between N and executive function remaining significant after FDR in the cross-sectional sample. N being a significant predictor of executive function and verbal memory in the longitudinal sample cemented N’s standing as the most influential Big Five variable on cognitive function. This connection between N and cognition was found in studies examining HC [39,40,41], and seems to be applicable to individuals with BD as well. A study by Chang et al. [52] yielded the same result. Similarly to HC, higher N might contribute to a higher frequency of unhealthy behaviors impacting long-term cognition among individuals with BD [44,45], which they are known to exhibit more than HC: they exercise less [66] and are more likely to smoke [67] and have a less healthy diet [68]. This is supported by a positive correlation between N and cardiovascular disease [32].

Executive function and verbal memory seem to be affected the most by the aforementioned correlates of N. Regarding the former, healthy older adults showed a negative association between N and executive function [69,70]. Fittingly, our previous research suggested a higher prevalence of metabolic syndrome in individuals with BD, which was found to be a risk factor for impairment of executive function [71]. In addition, damage to the left dorsolateral prefrontal cortex, which is important for executive function, was found to be associated with high N [72]. Verbal memory was predicted by N as well. Other studies found comparable results, suggesting a link between high N and worse memory recall in HC [65,73], as well as memory complaints [74] and greater decreases in verbal memory in older adults [75]. These results underline the importance of preventing cognitive decline in individuals with BD. For example, mindfulness training was found to reduce N [76], and a unified protocol designed specifically for decreasing N was successful [77]. Additionally, low C might contribute to cognitive decline [46], as affected individuals display less behavior contributing to the preservation of cognitive function [49].

C was an important predictor of executive function as well. Similarly, Sutin et al. [50] found better executive function to be predicted by higher C. High C has been linked to high achievement by students [78], better cognitive performance, greater pursuit of cognitive activities [79], and less cognitive decline [46]. As C was lower in the BD group compared to the norm sample, both C and executive function are important points to consider in therapy. In contrast to our result, other findings showed no association between C and cognitive function in HC [40] or individuals with BD [51]. Apart from the dissimilar study populations, these diverse results might be explained by different measures of executive function, which is a broad concept spanning several processes of cognition. For instance, Crow [40] used response to inhibition and sustained attention to represent executive function, while the current study used measures of cognitive flexibility, the TMT-B, and the interference trial of the Stroop test. In conclusion, the exact relationship between C and different categories of executive function in BD is still unclear. 

Attention and processing speed were solely predicted by age at t1. As the tests assessing these cognitive domains were mainly dependent on speed, it is not surprising that none of the Big Five were significant predictors. The progression of time, not personality, was the most impactful factor affecting the decline in attention and processing speed, which are known to decrease with age [80]. A possible neuronal correlate is fractional anisotropy of white matter [81]. Zhao et al. [82] have recently suggested that the deterioration of the glial structure disturbs the ratio between neural activity and the availability of oxygen.

Except for a significant correlation between O, attention, and processing speed prior to FDR, O unexpectedly did not correlate with or predict any variables related to cognition. High O has been consistently linked to higher cognitive abilities in HC [42,46] and has been found in individuals with BD as well [38], although results are scarce, and this connection in individuals with BD remains to be elucidated.

Finally, education was a significant predictor of executive function, but not verbal memory nor attention and processing speed at t2. Education has been identified as an important factor for the development of executive function in early childhood [83]; in turn, executive function is integral for pre-school learning [84]. Moreover, the quality of education during adulthood impacts executive function as well [85]. Many studies have investigated this reciprocal relationship in children, while older adults have been in the spotlight of a few studies showing the same results: a higher level of education is associated with better executive function [86,87], which seems to be true for middle-aged individuals with BD as well. Considering that executive function includes working memory, inhibition, and attention shifting [88], it is evident that these skills facilitate educational achievement, while education might help to hone them. This is supported by the fact that high conscientiousness is also a significant predictor of executive function, a trait that is linked to high achievement [78]. This is particularly relevant for individuals with BD, as those with depression have a lower education level than the general population [89]. Considering verbal memory, other studies have found a significant association with education [90,91]. As for attention and processing speed, education does not seem to hold as much importance as age, which we discussed previously.

The first limitation of the current study was the small sample size, especially in the longitudinal subsample, which might have led to an underestimation of the relationship between cognition and the Big Five. In this context, possible attrition bias due to high dropout rate might have occured, although there were no significant differences between t1 and t2 except age. Secondly, there was no control group with which to compare individuals with BD. The influence pattern of personality and cognition might be different in HC, and it would have been essential to compare both groups. Thirdly, not all cognitive tests were administered at t2 for the longitudinal sample, leading to mean scores of the three cognitive domains that were not as differentiated as the scores at t1. Fourthly, information on medication intake was assessed, but could not be included in the statistical analyses due to missing data. As cognitive performance might have been inhibited by the effect of certain medication, it would have been important to include this variable as a covariate. Fifthly, the time span between t1 and t2 was heterogenous in the cross-sectional sample, although there were no outliers, and might have influenced the results. Finally, subscores of the Big Five were not administered and might provide a more differentiated picture of the relationship between cognition and personality. Related to this are different measures of executive function, which might lead to differing results and should be examined in more detail.

## 5. Conclusions

Individuals with BD have lower N, E, and C than HC. High N has a strong association with worse executive function and verbal memory, while C is positively correlated with executive function. While the Big Five might not strongly impact cognitive function over the span of several years, they are, nevertheless, significant predictors which might take effect over a longer period of time. Future longitudinal studies should include a higher number of participants as well as more time in between points of measurement.

## Figures and Tables

**Table 1 brainsci-13-00773-t001:** Correlations between the Big Five and cognitive function in healthy controls.

Big Five Factor	Correlation with High Cognitive Function
Neuroticism	negative
Openness	positive
Conscientiousness	positive
Extraversion	positive
Agreeableness	unclear

**Table 2 brainsci-13-00773-t002:** Sociodemographic information, Big Five, and cognitive test scores of the cross-sectional sample of individuals with bipolar disorder.

Variables	Cross-Sectional Sample (n = 129) M (±SD)
Age	43.45 (13.25)
Sex (n)	
Male	64 (49.6%)
Female	65 (50.4%)
Type of bipolar disorder	
Type 1	87 (67.4%)
Type 2	42 (32.6%)
HAMD	5.02 (4.03)
YMRS	1.13 (2.62)
Big Five	
Openness	29.43 (6.74)
Conscientiousness	31.11 (7.53)
Extraversion	25.43 (6.94)
Agreeableness	30.38 (6.37)
Neuroticism	27.36 (8.68)
Cognition	
TMT-A	34.71 (15.04)
TMT-B	80.64 (41.57)
d2-R	146.76 (43.26)
Stroop color word reading	31.53 (5.50)
Stroop color naming	48.92 (8.31)
Stroop interference	81.11 (20.45)
CVLT trial 1-5	53.09 (12.84)
CVLT short delay free recall	10.64 (3.55)
CVLT short delay cued recall	11.67 (3.22)
CVLT long delay free recall	11.56 (3.54)
CVLT long delay cued recall	11.97 (3.28)

Note. HAMD = Hamilton Depression Rating Scale; YMRS = Young Mania Rating Scale; TMT = Trail Making Test, d2-R = d2-Test Revised; CVLT = California Verbal Learning Test.

**Table 3 brainsci-13-00773-t003:** Sociodemographic information, Big Five, and cognitive test scores of the cross-sectional sample of individuals with bipolar disorder.

Variables	Longitudinal Sample (n = 35)				
	t1 (M ± SD)	t2 (M ± SD)	Statistics	*p*	η^2^
Age	43.99 (14.65)	46.14 (14.67)	t = −9.22	<0.001	
Sex (n)					
Male	16 (45.7%)				
Female	19 (54.3%)				
Type of bipolar disorder					
Type 1	23 (65.7%)				
Type 2	12 (34.3%)				
HAMD	3.63 (3.24)	3.17 (3.09)	t = 0.20	0.846	
YMRS	1.88 (3.98)	1.46 (3.15)	Z = −0.41	0.682	
Time difference between t1 and t2 (days)		959.03 (462.92)			
Big Five					
Openness	29.49 (6.46)				
Conscientiousness	31.43 (7.44)				
Extraversion	25.51 (6.46)				
Agreeableness	30.11 (6.12)				
Neuroticism	26.97 (8.06)				
Cognition					
TMT-A	34.82 (10.78)	28.18 (12.89)	F = 1.53	0.228	0.03
TMT-B	83.30 (46.30)	63.19 (31.31)	F = 1.78	0.195	0.05
Stroop color word reading	32.42 (5.29)	32.64 (6.66)	F = 1.21	0.282	0.00
Stroop color naming	50.52 (9.10)	50.26 (10.66)	F = 1.20	0.283	0.01
Stroop interference	84.21 (20.03)	81.57 (23.71)	F = 1.15	0.294	0.03
CVLT trial 1-5	52.51 (13.92)	51.05 (11.28)	F = 0.10	0.757	0.01

Note. Cognitive differences were calculated using repeated-measures multivariate analyses of covariance (controlled for age, sex, education, BDI, illness duration, and time difference) testing differences between t1 and t2. HAMD = Hamilton Depression Rating Scale; YMRS = Young Mania Rating Scale; TMT = Trail Making Test, d2-R = d2-Test Revised; CVLT = California Verbal Learning Test.

**Table 4 brainsci-13-00773-t004:** Correlations between the Big Five and cognitive function at t1 in the cross-sectional sample of individuals with bipolar disorder (n = 129).

Big Five	Cognitive Function					
	Executive Function ^a^		Verbal Memory ^b^		Attention and Processing Speed ^c^	
	r	*p*	r	*p*	r	*p*
Openness	0.15	0.112	0.04	0.687	0.22	0.020
Conscientiousness	0.02	0.827	−0.08	0.396	−0.01	0.904
Extraversion	0.04	0.691	0.11	0.269	0.13	0.168
Agreeableness	−0.01	0.942	−0.03	0.792	0.07	0.448
Neuroticism	**−0.28**	**0.003**	0.09	0.927	−0.18	0.058

Note. BD = bipolar disorder; ^a^ sum score of the Trail Making Test (TMT) part B and the interference trial of J. Stroop’s Color and Word Interference Test; ^b^ sum score of California Verbal Learning Test (CVLT) trial 1-5, CVLT short delay free recall, CVLT long delay free recall, CVLT short delay cued recall, and CVLT long delay cued recall; ^c^ sum score of d2 Test of Attention Revised, Stroop’s word-reading and color-naming trials, and TMT part A. Results remaining significant after the employment of False Discovery Rate (FDR) are marked in bold letters. Covariates included age, sex, education, and BDI.

**Table 5 brainsci-13-00773-t005:** Correlations between the Big Five at t1 and the cognitive function differences between t1 and t2 in the longitudinal sample of individuals with bipolar disorder (n = 35).

Big Five	Cognitive Function Differences between t1 and t2					
	Executive Function ^a^		Verbal Memory ^b^		Attention and Processing Speed ^c^	
	r	*p*	r	*p*	r	*p*
Openness	0.01	0.962	−0.07	0.751	0.10	0.966
Conscientiousness	−0.30	0.145	−0.14	0.491	−0.09	0.683
Extraversion	0.15	0.482	−0.02	0.916	0.20	0.338
Agreeableness	−0.19	0.368	−0.38	0.059	−0.19	0.376
Neuroticism	−0.02	0.929	0.30	0.141	−0.05	0.823

Note. BD = bipolar disorder; ^a^ difference between sum scores at t1 and t2 according to the Trail Making Test (TMT) part B and the interference trial of J. Stroop’s Color and Word Interference Test; ^b^ difference between California Verbal Learning Test (CVLT) trial 1-5 scores at t1 and t2; ^c^ difference between sum scores at t1 and t2 according to Stroop’s word-reading and color-naming trials as well as TMT part A. A false discovery rate (FDR) correction for multiple comparisons was used. Covariates included age, sex, education, BDI, illness duration, and time difference between t1 and t2.

**Table 6 brainsci-13-00773-t006:** Multiple hierarchical regression analysis predicting executive function at t2.

Model	Variables	Executive Function ^a^		
		β	t	*p*
Model 1	Age	−0.24	−0.96	0.347
	Sex	0.16	0.92	0.366
	Education	0.43	2.17	**0.041**
	Time difference t1–t2	0.13	0.76	0.458
	BDI	−0.14	−0.82	0.418
	Illness duration	0.08	0.38	0.710
Model 2	Age	−0.12	0.58	0.576
	Sex	0.35	1.89	0.074
	Education	0.41	2.26	**0.036**
	Time difference t1–t2	0.56	2.60	**0.018**
	BDI	0.38	1.66	0.113
	Illness duration	0.09	0.47	0.644
	Openness	−0.05	−0.32	0.753
	Conscientiousness	0.48	2.55	**0.019**
	Extraversion	−0.30	−1.44	0.166
	Agreeableness	−0.09	−0.51	0.616
	Neuroticism	−0.82	−3.30	**0.004**

Note. Model 1: R^2^ = 0.38, R^2^ corr. = 0.23, SE = 1.48. Model 2: R^2^ = 0.65, R^2^ corr. = 0.45, SE = 1.25. Significant *p*-values (*p* < 0.05) are written in bold. ^a^ Sum score of the Trail Making Test part B and the interference trial of J. Stroop’s Color and Word Interference Test.

**Table 7 brainsci-13-00773-t007:** Multiple hierarchical regression analysis predicting verbal memory at t2.

Model	Variables	Verbal Memory ^a^		
		β	t	*p*
Model 1	Age	−0.62	−2.68	**0.013**
	Sex	0.17	1.05	0.303
	Education	0.11	0.56	0.581
	Time difference t1–t2	0.09	0.54	0.594
	BDI	0.11	0.67	0.512
	Illness duration	0.13	0.62	0.538
Model 2	Age	−0.55	−2.66	**0.016**
	Sex	0.33	1.87	0.078
	Education	−0.02	−0.14	0.890
	Time difference t1–t2	0.23	1.11	0.280
	BDI	0.45	2.03	0.057
	Illness duration	0.22	1.13	0.273
	Openness	0.16	1.08	0.293
	Conscientiousness	0.13	0.73	0.477
	Extraversion	−0.27	−1.38	0.184
	Agreeableness	0.17	1.08	0.295
	Neuroticism	−0.68	−2.84	**0.011**

Note. Model 1: R^2^ = 0.45, R^2^ corr. = 0.31, SE = 0.81. Model 2: R^2^ = 0.68, R^2^ corr. = 0.49, SE = 0.70. Significant *p*-values (*p* < 0.05) are written in bold. ^a^ Score on the California Verbal Learning Test (CVLT) trial 1-5.

**Table 8 brainsci-13-00773-t008:** Multiple hierarchical regression analysis predicting attention and processing speed at t2.

Model	Variables	Attention and Processing Speed ^a^		
		β	t	*p*
Model 1	Age	−0.45	−2.05	0.051
	Sex	0.06	0.42	0.680
	Education	0.23	1.33	0.197
	Time difference t1–t2	0.15	0.10	0.349
	BDI	−0.27	−1.71	0.099
	Illness duration	0.01	0.04	0.970
Model 2	Age	−0.39	−2.10	**0.049**
	Sex	0.26	1.24	0.229
	Education	0.20	1.07	0.300
	Time difference t1–t2	0.36	1.60	0.127
	BDI	0.01	0.03	0.977
	Illness duration	0.02	0.10	0.922
	Openness	0.00	0.00	1.00
	Conscientiousness	0.24	1.23	0.235
	Extraversion	−0.29	−1.34	0.195
	Agreeableness	−0.05	−0.27	0.791
	Neuroticism	−0.52	−1.02	0.057

Note. Model 1: R^2^ = 0.52, R^2^ corr. = 0.40, SE = 1.85; Model 2: R^2^ = 0.60, R^2^ corr. = 0.43, SE = 1.84. Significant *p*-values (*p* < 0.05) are written in bold. ^a^ Sum score of Stroop word-reading and color-naming trials of the Color and Word Interference Test by J. Stroop and the Trail Making Test (TMT), part A.

## Data Availability

The data presented in this study are available upon request from the corresponding author. The data are not publicly available due to privacy or ethical restrictions.

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
