# Peer review of "The Big Five as Predictors of Cognitive Function in Individuals with Bipolar Disorder"

_brainsci, 2023, doi:10.3390/brainsci13050773_

Round 1
Reviewer 1 Report
Comments and Suggestions for Authors
The manuscript reports an interesting study about the relationships between the Big Five and cognitive functioning in bipolar disorder. The manuscript reports a partial sample of patients of a longitudinal ongoing study. The paper is well-written, and the methods are clear. However, I have some concerns about specific points and the authors should address:
- your samples are quite small for an ongoing study. Is there a specific reason (for example refusing to participate) for a small longitudinal sample (a quarter of the cross-sectional)?
- Have you evaluated the sample sizes required before the study? At least, can you evaluate your sensitivity now?
- The absence of a control group is a serious limit regards the ability to define these characteristics as linked to bipolar disorder. The authors have already reported this aspect in the limits, but I think they should reduce the emphasis of some parts of their paper.
- Please, specify the specific diagnoses of the included sample (type 1 or 2?). Moreover, how did you perform the diagnosis? Who performed the diagnosis?
- What about medication? Have you evaluated the possible effects on cognitive performances?
- Why did you exclude some parts from the longitudinal analysis?
- What about the duration of the disorder? There is a linkage between neuro-progression and cognitive functioning in bipolar disorder.
Author Response
Dear Reviewer 1,
Thank you for reviewing our manuscript. With your help, we were able to make significant improvements. Attached, you can find the new version.
Best wishes,
the authors

Reviewer 2 Report
Comments and Suggestions for Authors
Thank you for giving me the opportunity to review this manuscript.
1) Please specify only one study design. This study must not contain both cross-sectional and longitudinal studies. I think this study must not be a cohort study because of too small sample size. This study should be an only cross-sectional study. Please describe that this study was a cross-sectional study in the title, abstract, and the methods.
2) Please delete the sentences of longitudinal methods and results. For example, please delete the sentences of the followings, and the results of Multiple hierarchical regression analyses.
a) "The longitudinal sample of 35 individuals with BD additionally included assessments from a second visit yielding complete datasets (t2), taking place 345 days to 2106 days after the first visit"
b) "For the longitudinal sample, the same procedure was repeated, however, not all cognitive test scores could be used for the creation of domains scores, as they were not assessed at t2. For memory, only the score for recall trials 1-5 was included in the sum score. The sum score of attention and processing speed did not include the d2-R test, which was accounted for when calculating the sum score and comparing both samples."
c) "To compare t1 and t2 of the longitudinal sample, t-tests were used for age, HAMD, and a Wilcoxon Test was used for YMRS due to outliers. A repeated measures multivariate analysis of variance (MANCOVA) was employed to test for differences in cognition between both time points. Only the variables assessed at both time points were included: TMT A, TMT B, recall trials 1-5 of the CVLT, and the Stroop test. Covariates included age, sex, education, BDI, and time difference between t1 and t2. "
d) "Furthermore, the same analyses were performed for correlating each of the Big Five assessed at t1 and cognition at t1, t2, and the difference between the three cognitive functions measured at t2 as compared to t1. In addition to the formerly mentioned covariates, the time difference between t1 and t2 was used for the latter analysis.
3) Please clearly define all predictors, potential confounders, and effect modifiers. Please describe any efforts to address potential sources of bias.
4) Please explain how the sample size was arrived at. Please describe analytical methods taking account of sampling strategy.
5) Please describe all statistical methods, including those used to control for confounding.
6) Please explain how missing data was addressed. Please indicate number of participants with missing data for each variable of interest.
7) Please describe any information on exposures and potential confounders.
8) The authors described "Moreover, social [17] and emotional cognition can be impaired as well, specifically emotion recognition [18] and emotion regulation in social situations [19]." I think that emotional cognition and emotional recognition were specific domains of social cognition. Please clarify what is neurocognition, what is social cognition, what is the domains of the neurocognition, and what is the domains of social cognition.
9) The authors described that " As there is currently a lack of studies investigating personality and cognitive functioning in BD, the aim of this study was 1. to investigate cross-sectional correlations between the Big Five as well as executive function, verbal memory, and attention and processing speed", while "Another study suggested a negative correlation between N and reaction time in the Affective Go/No-Go paradigm with a bias towards affective stimuli, which was not significant in HC and might suggest a higher receptivity to emotional stimuli in individuals with BD [52]." I think that "a lack of studies investigating personality and cognitive functioning in BD" was wrong. Reaction time in the Affective Go/No-Go paradigm was used to assess processing speed (a neurocognitive domain). Please describe more clearly and more in detail what have been found in the previous studies, and what is the novelty of this study as the cross sectional study.
Author Response
Dear Reviewer 2,
Thank you for reviewing our manuscript. Your suggestions were helpful and we believe that we were able to make significant improvements. Attached you can find the new version.
Best wishes,
the authors

Round 2
Reviewer 1 Report
Comments and Suggestions for Authors
I think the authors have made some significant changes. I think the paper reports preliminary data due to the limits reported in the manuscript.
Reviewer 2 Report
Comments and Suggestions for Authors
This study must not be any longitudinal design.
The sample size was too small to investigate anything by this longitudinal design.